# The Temporal Distribution of the Host Rocks to Gold, the Archean Witwatersrand Basin, South Africa

Neil Phillips [1,2,*], Julian Vearncombe [1,3], Dave Craw [4] and Arthur Day [5]

1 Earth Sciences, Stellenbosch University, Stellenbosch 7602, South Africa; julian@vearncombe.com
2 School of Geography, Earth and Atmospheric Sciences, University of Melbourne, P.O. Box 3, Central Park, Melbourne, VIC 3145, Australia
3 Effective Investments Pty Ltd., P.O. Box 675, South Perth, WA 6951, Australia
4 Geology Department, University of Otago, Dunedin 9010, New Zealand; dave.craw@otago.ac.nz
5 Independent Consultant, P.O. Box 533, Corrimal, NSW 2518, Australia; arthurday2000@yahoo.com
* Correspondence: neil@phillipsgold.com.au

**Abstract:** The hosts to gold around the Witwatersrand Basin span over 400 my, through 14 km of stratigraphy in a variety of host rocks and in tectonic settings that include periods of rifting, thermal subsidence, foreland basin, flood basalt outpouring, graben development, and further thermal subsidence. A geological model that assumes placer processes to explain this diversity implies a super-long-lived and special source of the detrital gold, transport, and highly effective sorting processes over a time span of 400 my. There is no evidence of a special source and sorting over such a long time period. In the Phanerozoic, this would be equivalent to the special source and sorting processes operating continually over an equivalent period of geological time spanning from the Devonian up until the present day; this is as yet recognised nowhere else on the planet. With regard to the geological model that assumes a placer process, this is untenable because of these scientific shortcomings and its lack of success in exploration. A better use of funds may be to consider alternative approaches and epigenetic models in exploration.

**Keywords:** gold; Witwatersrand; unconformity; basin scale; Archean; alteration

## 1. Introduction

Geological models inform mineral exploration with the aim of aiding discoveries. Geological models morph into exploration methods, dictating which ground to acquire, how to sample, how and where to drill, and resource assessment [1]. Geological models can be descriptive, genetic, or a combination of both. In their simplest form, the *descriptive* model can involve pattern recognition observing some aspect of existing mineralisation, such as gold occurring with pyrite and quartz veins, and then looking for repetitions. The *genetic* model involves determining how deposits have formed and using these ideas as a prediction tool in exploration. The expectation is not for perfection in a model, but to use any model with caution and, in a feedback loop, ask whether the model fits the existing data and aids discovery. It is common to combine descriptive and genetic models to construct improved exploration that can inform area selection and appropriate testing methods.

The Witwatersrand goldfields of South Africa have a long history of exploration, illustrating both the descriptive and the genetic components. Some of this exploration has been spectacularly successful, whereas some has been disappointing. The initial discovery of Witwatersrand gold was made by George Harrison at Langlaagte within present day Johannesburg when he noticed gold grains in a quartz pebble conglomerate in early 1886. Within months, that descriptive model of a conglomerate host rock had been followed along strike east and west for tens of kilometres to reveal what became the Central Rand goldfield (290 Moz Au approximate all-time production; Figure 1). The West Rand (130 Moz) and East Rand (280 Moz) goldfields followed the Central Rand success before

1900 after the resolution of some structural breaks in the continuity of the conglomerate and observation of other conglomerate horizons. Major discoveries after 1930 required a change of focus away from the conglomerates to a model of reef packages on major unconformities, comprising carbon seam, clean well sorted sandstone, and shale, as well as conglomerate, or subsets of this sedimentary package. The discoveries of Carletonville (290 Moz), Welkom (350 Moz), Vaal Reef at Klerksdorp (230 Moz for the goldfield), and Evander (60 Moz) goldfields were under cover and required an understanding of the stratigraphy, tracing it with geophysics, and utilising the unconformities and reef packages in a process that would not have been possible in 1886. From the 1960s, a geological model that assumes a placer process for the Witwatersrand gold was extensively used in research and genetic-based exploration in and around the goldfields, but the period after the Evander goldfield discovery of 1951 through until the 2020s has failed to reveal any new goldfields [2]. To understand and explain this on-going 70-year period in which no new goldfield has been discovered, it is important to review the placer model upon which that exploration was based, and especially to re-examine the currently available descriptive geology and whether it still supports the placer model.

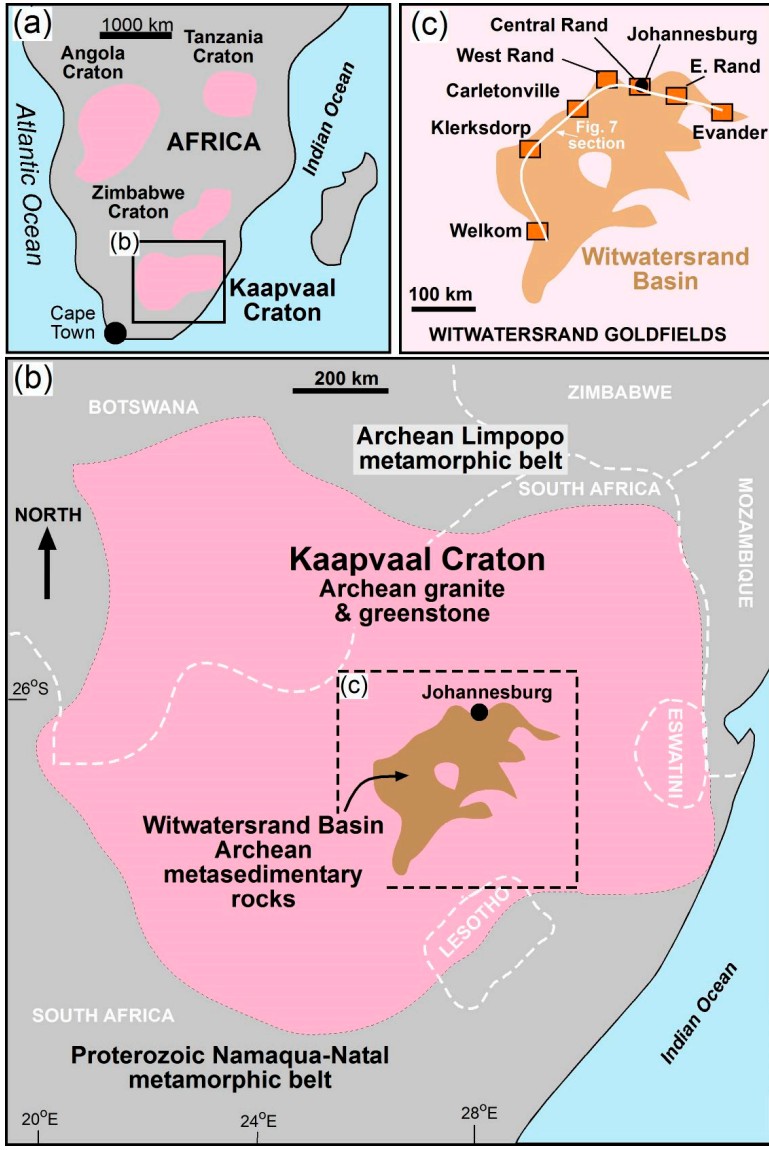

**Figure 1.** (**a**) Map of southern Africa highlighting the Archean Kaapvaal Craton (**b**), the Witwatersrand Basin, Johannesburg, and the seven major goldfields. In (**c**), the white line connecting the goldfields shows the extent of the Witwatersrand gold reefs.

We suggest that there is no descriptive information that is more important for geological models than the distribution of gold at all scales. This includes gold at economic and at anomalous levels, and on microscopic, mesoscopic, goldfield, and whole-basin scales. Here we investigate the distribution of gold at the economic level and at the whole-basin scale by noting where it has been mined, and specifically the age of host strata. Our earlier studies have focused on economic gold at the goldfield scale [3] and anomalous gold at the mesoscopic to whole-basin scale [4–6].

## 2. Approach and Methods

This study investigates economic gold in four Archean supracrustal sequences in northern South Africa (Figure 2). Three are known over several hundred kilometres, whereas the distribution of the lowest Dominion Group is poorly known but spans at least one hundred kilometres. Descriptions of these four supracrustal sequences is readily available in Geology of South Africa [7], which includes relevant chapters on the Dominion Group [8], Witwatersrand Supergroup [9], Ventersdorp Supergroup [10], and Transvaal Supergroup [11,12]. The Witwatersrand Supergroup has been reviewed with the emphasis on the seven major goldfields and their main reef groups by [13–15].

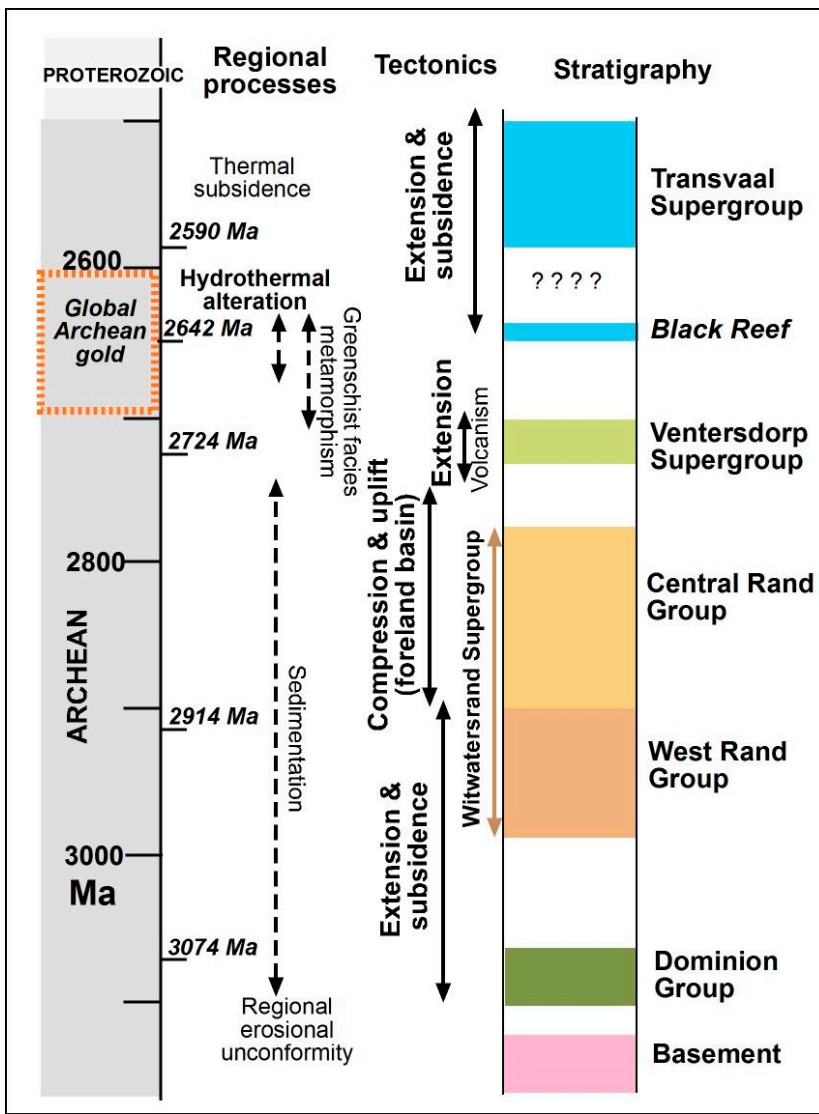

**Figure 2.** Time-based stratigraphic column of the middle and late Archean from 3100 to 2500 Ma showing successor basins and significant geological events. The stratigraphic correlation of Black Reef mineralisation in Witwatersrand gold mines is uncertain in places (shown by ????).

Syntheses by [16] are compiled and written with a global perspective; the author was able to utilise his extensive access to mining personnel, data, and mines which have been closed now for decades. The summaries of [17], the independent Chamber of Mines production figures, Council of Geoscience in Pretoria, more recent reports by mine geologists, and the authors' own research have been used to supplement and confirm the broad tenets of Pretorius's synthesis that we use here.

Our aim is to synthesise the various South African examples of economic gold in Precambrian conglomerate, particularly quartz pebble conglomerates and related strata. Of interest is their age, host package, stratigraphic position, tectonic setting at sedimentation, geographic setting today, deformation, and relationship to large-scale alteration. We test the placer model for gold against these observations and descriptive detail focusing on the source of detrital gold and the sedimentary sorting of those grains. Some important information resides in reports from the early 20th century made by geologists on mines that are now closed, and their works are rarely referred to this century. A key question is whether a geological model assuming a placer origin remains fit for purpose today, given what is known about the Witwatersrand in the 2020s.

## 3. Global Importance of Gold in Precambrian Quartz Pebble Conglomerates

In a major review in the 75th anniversary volume of Economic Geology, [16] discusses the global group of gold and/or uranium deposits in quartz pebble conglomerates. He lists seven sequences that meet his criterion of having "provided ore for large-scale and long-term mining operations" (pp. 119–120), which are as follows:

- upper Witwatersrand Johannesburg subgroup;
- upper Witwatersrand Turffontein subgroup;
- Tarkwa Supergroup in Ghana;
- Blind River in Canada;
- Black Reef in South Africa;
- lower Witwatersrand Government subgroup in South Africa;
- Dominion Group in South Africa.

In this major review of auriferous conglomerates globally, Pretorius included over twenty further examples, including the Pongola and Uitkyk conglomerates in South Africa, but none met his criteria of mining duration and scale.

Tarkwa in Ghana, Blind River in Canada, and the upper Witwatersrand are all well known for their gold and/or uranium in conglomerate. However, what will surprise many, particularly in South Africa, is the global importance of the three further South African examples of conglomerate-hosted gold deposits (Figure 3). These would rarely be regarded as world-class examples in South Africa, simply because they appeared relatively modest in size and have been overwhelmed by the massive production that has come from the upper Witwatersrand (Central Rand Group).

The coverage of these deposits by Pretorius is slightly dated in lacking examples from the 1980s onwards; he also worked within an earlier stratigraphic nomenclature which used some names that differ from the current stratigraphy [9]. Another limitation is that the Dominion, Government, and Black Reef mines cannot be quantified in terms of modern Reserves and Resources (their mining predated these measures), and all-time production figures are incomplete. However, their importance derives from quantitative comments on the scale of mining operations, the importance of their contribution to the early Witwatersrand gold production, and the value of Pretorius's syntheses and judgements. We estimate an all-time production of at least 20 Moz from the combined Dominion and West Rand Groups, compatible with other estimates.

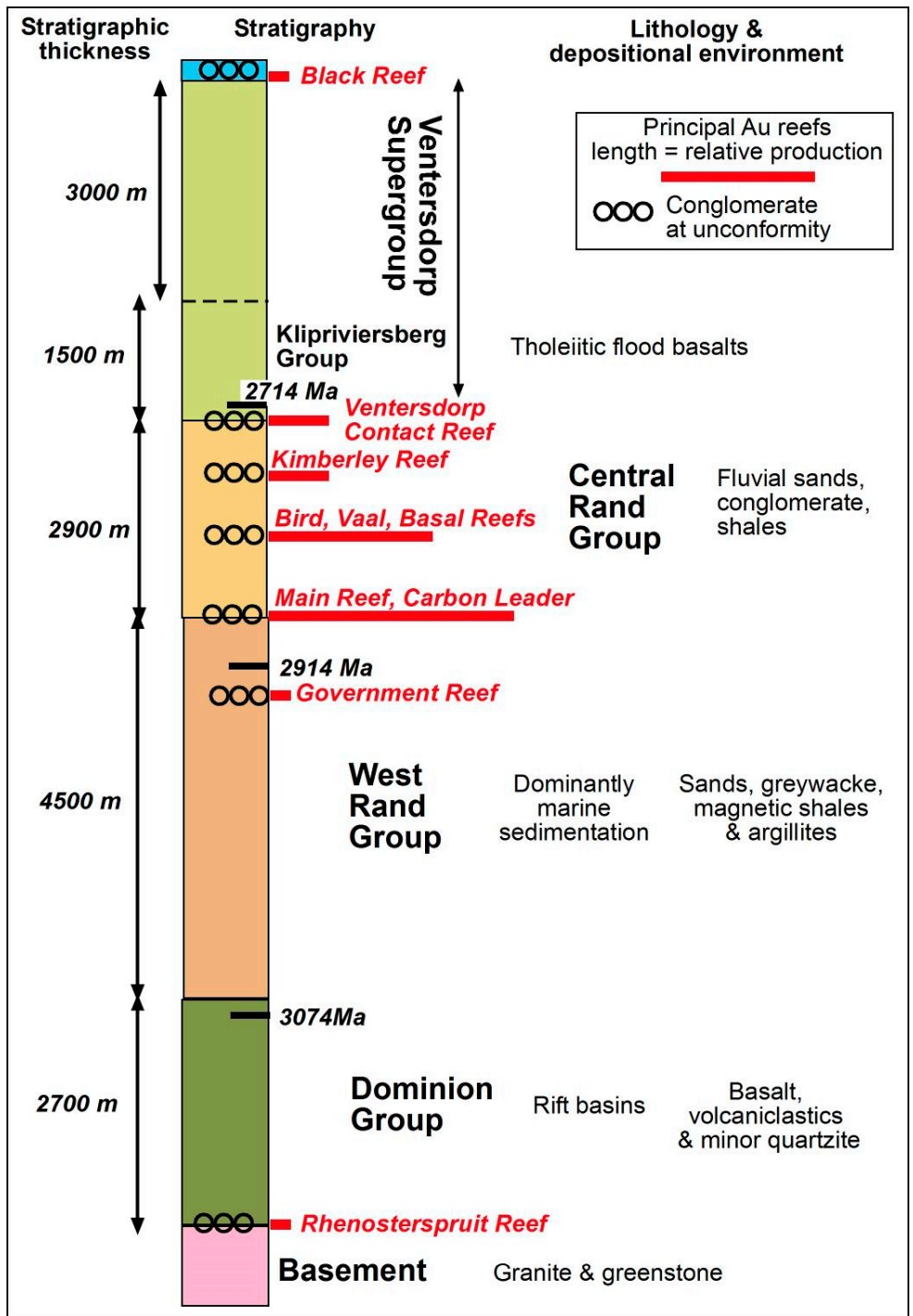

**Figure 3.** Stratigraphic column showing approximate thickness of various Archean successor basins with their main rocks and reef groups. The red bars for the major gold reef groups show their relative productions. The Dominion, West Rand Group, and Black Reef production bars are approximate only, based on various sources. The OOO symbol denotes reef packages.

## 4. Geological Setting of Reef Packages and Reef Groups

South Africa has a number of Precambrian successor basins on the granite–greenstones of the Kaapvaal Craton [7]. Major components of the Kaapvaal Craton are:

- Transvaal (Black Reef at the base is dated at 2642 +/− 2 Ma);
- Ventersdorp (base is 2729 +/− 19 Ma);
- Witwatersrand;

○     Central Rand Group (2902 Ma base to 2780 Ma top);
○     West Rand Group (Crown basalt towards the top is 2914 +/−8 Ma);
●     Dominion (the top is dated at 3074 +/− 6 Ma);
●     Granite–greenstone basement.

The Witwatersrand Supergroup is a ~7.5 km thick sedimentary basin with minimal igneous rocks but with Archean greenstone strata above and below. Within the Supergroup are multiple unconformities. These extensive planar surfaces can be 100 to 400 km$^2$ in area and are overlain immediately by distinctive reef packages, with many containing economic gold. Most Witwatersrand gold is immediately above planar unconformity surfaces and not restricted to, or concentrated in, erosion channels that are incised through the reefs. However, in modern alluvial fans or braided streams, gold is almost entirely in erosion channels on a smaller scale than the Witwatersrand reef packages and not spread across the extensive planar unconformities [3].

### 4.1. Dominion Group

The Dominion Group unconformably overlies the Archean granite basement and is dated at 3074 ± 6 Ma (single zircon U-Pb SHRIMP [18]). The Dominion Group is unconformably overlain by the Witwatersrand Supergroup and thus places a maximum age on the latter [19].

The Dominion Group conglomerates contain pyrite, gold, altered detrital Ti-magnetite, monazite, cassiterite, carbon, uraninite, coffinite, and ilmenite. Gold was mined from 1888 and was significant enough to cause ([20], p. 181) to report that "the Dominion reefs made a worthwhile contribution to gold production from Klerksdorp Field" (keeping in mind that this refers to the early days of the Klerksdorp goldfield prior to discovery of the Vaal Reef).

Outcrops of the Dominion Group are restricted to Ottosdal and Vredefort, approximately 50 km from Klerksdorp (to the west and east, respectively; [9]4). The Rhenosterspruit Formation at the base of the Dominion Group unconformably overlies granite and comprises 120 m of dominantly clastic metasedimentary rocks. The Rhenosterspruit Formation includes arkose, sandstone, grit, and two mineralised conglomerate horizons. The Lower Reef is essentially on the granitic basement and separated by 10–20 m of quartzite from the Upper Reef that is up to a metre thick [21]. The remainder of the Dominion Group is ~2.5 km of basalt and andesite [8,22].

Interest in uranium following the Second World War meant focus on the more uranium-rich Upper Reef in the Ottosdal area, which produced 1500 t $U_3O_8$ and included ores with some of the highest U values in the Witwatersrand Basin of 500 g/t $U_3O_8$ and, locally, one percent uranium ([21,23], p. 191).

Demonstrating significant epigenetic alteration, there are multiple intervals of pyrophyllite-rich rock in the Dominion Group and these have been mined commercially for the pyrophyllite ([22]; Figure 2). The abundant pyrophyllite has gained some scientific interest given that this sheet silicate mineral is common in Witwatersrand reefs in all the major goldfields in close association with gold ores [24,25], where it is interpreted as a product of regional-scale alteration.

### 4.2. Witwatersrand Supergroup

The Witwatersrand Supergroup lies upon ~3.0 Ga granites of the Kaapvaal Craton (Figure 4) and the Dominion Group and is itself overlain by ~2.7 Ga metabasaltic lavas [18]. The geochronology is compiled by [26], hence, detail is not tabulated here and is used in simplified form.

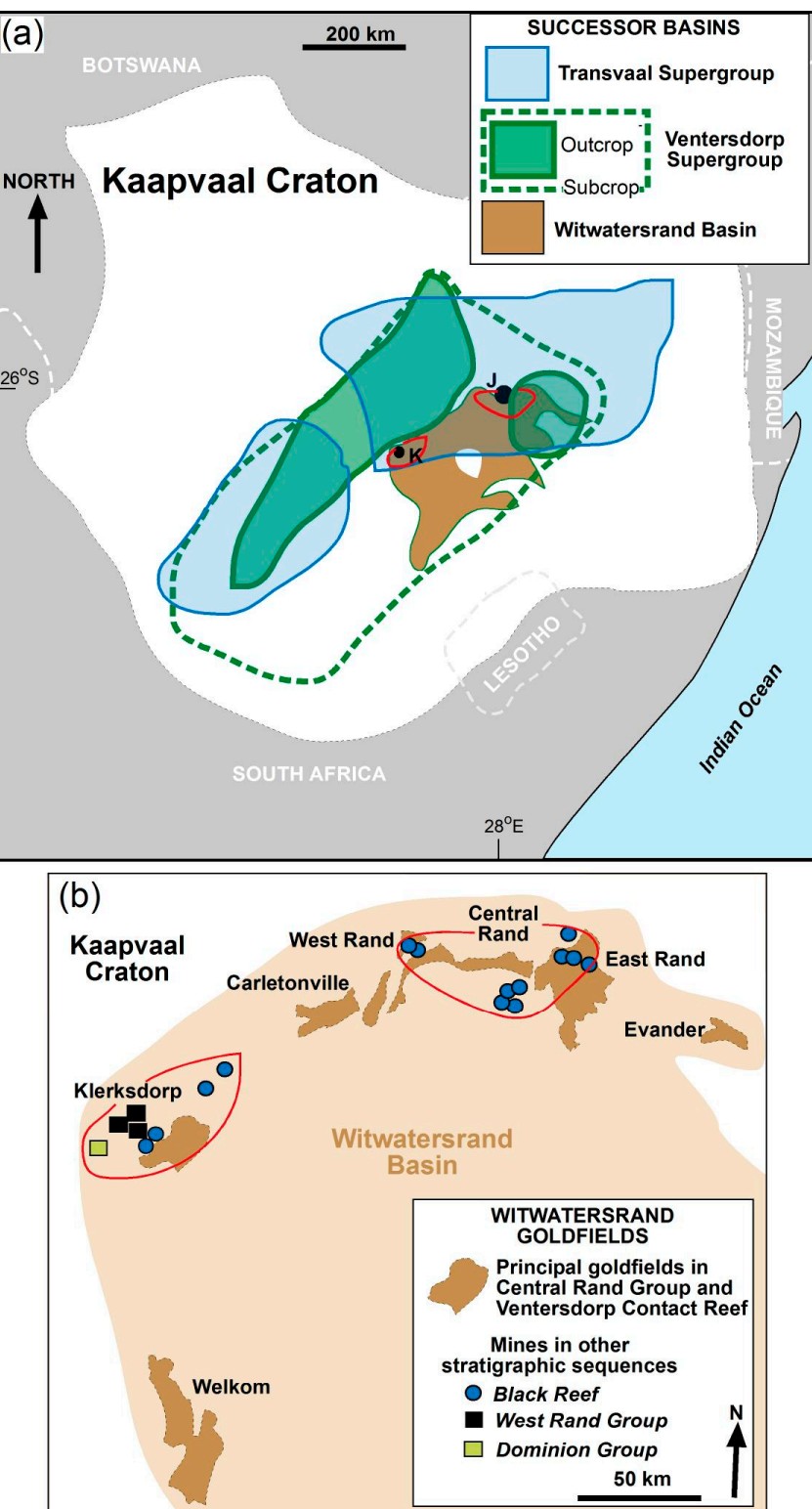

**Figure 4.** (**a**) Map of the Kaapvaal Craton (white) showing the approximate outcrop and subsurface extent of the Ventersdorp Supergroup, and extent of the Transvaal Supergroup including the western part, which includes Griqualand. The significant auriferous conglomerate reefs are concentrated between Klerksdorp (K) and Johannesburg (J) and within the two areas outlined in red. (**b**) The major goldfields reflect where gold has been mined from the Central Rand Group. The most important mines in the Dominion and West Rand Groups and Black Reef conglomerate are immediately adjacent to the major goldfields. The Black Reef Formation at the base of the Transvaal has been explored for hundreds of kilometres without success in discovering sustainable new mines.

The Witwatersrand Basin is divided into a lower West Rand Group dominated by ~4.5 km of fine- to medium-grained clastic metasedimentary rocks, and an upper Central Rand Group of ~3 km of medium- to coarse-grained clastic rocks with minor shale units [9]. This overall coarsening upward pattern has been interpreted as a trend from marine to more non-marine conditions. The Crown and Bird metabasaltic units are subordinate but useful in correlation, and the former provides an important age for the top part of the West Rand Group.

The Witwatersrand Supergroup is best known for outcrops and mine exposures for 300 km from Evander to the East Rand, Johannesburg, Carletonville, Klerksdorp, and Welkom. The lower part of the Supergroup is widespread and probably covers a large proportion of the Kaapvaal Craton [9]. The overall rock types vary from a shale–sandstone dominated lower part to a sandstone–conglomerate dominated upper part, with numerous unconformity surfaces mapped in mines and drill core. These assist basin-wide correlation [27] and include an unconformity separating the lower and upper Witwatersrand successions (Figure 5).

### 4.2.1. Lower Witwatersrand (West Rand Group)

The lowermost unit of the West Rand Group is younger than $2985 \pm 14$ Ma [28] and the Crown basalt near the top is $2914 \pm 8$ Ma [18]. The West Rand Group (lower Witwatersrand) has equal proportions of shelf mudstone and arenite of mainly marine origin. Laterally persistent magnetic shales and banded iron formations occur throughout the West Rand Group as 13 discrete units for over 200 km from Klerksdorp to the South Rand goldfield [29]. Gold and uraninite have been mined from several reefs in the West Rand Group, but in much smaller amounts than from the Central Rand Group.

It is worthwhile observing that the banded iron formations are highly deformed, such that they are locally known as the Contorted Bed, which crops out beside the campus of the University of the Witwatersrand [30].

There are several conglomerate bands, mostly less than one metre thick, on unconformity surfaces throughout the lower Witwatersrand and, although they have some gold, they are generally uneconomic. Some exceptions here are six reefs towards the top of the Government Subgroup, and the Government reef has been correlated over 300 km, including the South Rand, Klerksdorp, and Vredefort areas [31]. It varies from a few cm to over 2 m thick and comprises pebbles generally less than 5 cm diameter.

The Buffelsdoorn mine, 20 km NE of Klerksdorp, was very significant in the early days, with 170 stamp batteries (more than any mines in Johannesburg). It extracted gold from reefs near the top of the Government Subgroup and base of the Jeppestown Subgroup ([20], p. 185). The Babrosco and Afrikander Lease operations mined reefs at the base of the Jeppestown Subgroup with success and both operated for several decades. An important marker in the reef packages has been the Marble Quartzite immediately above conglomerate [32], akin to the reef packages of the upper Witwatersrand (CRG).

Separately to these examples in the Klerksdorp district, there has been mining of gold from the lower Witwatersrand in the Central Rand goldfield, including the auriferous quartz veins of the Confidence Reef in Roodepoort, 18 km NW of Johannesburg, which is 1–2 km north and removed from the Central Rand Group (Figure 4). The very small Edenskop mine in the South Rand goldfield produced ~3000 oz Au from the Coronation reef. There are several other very small mines in the lower Witwatersrand adjacent to the Vredefort Dome that appear to be within the regolith (weathered zone).

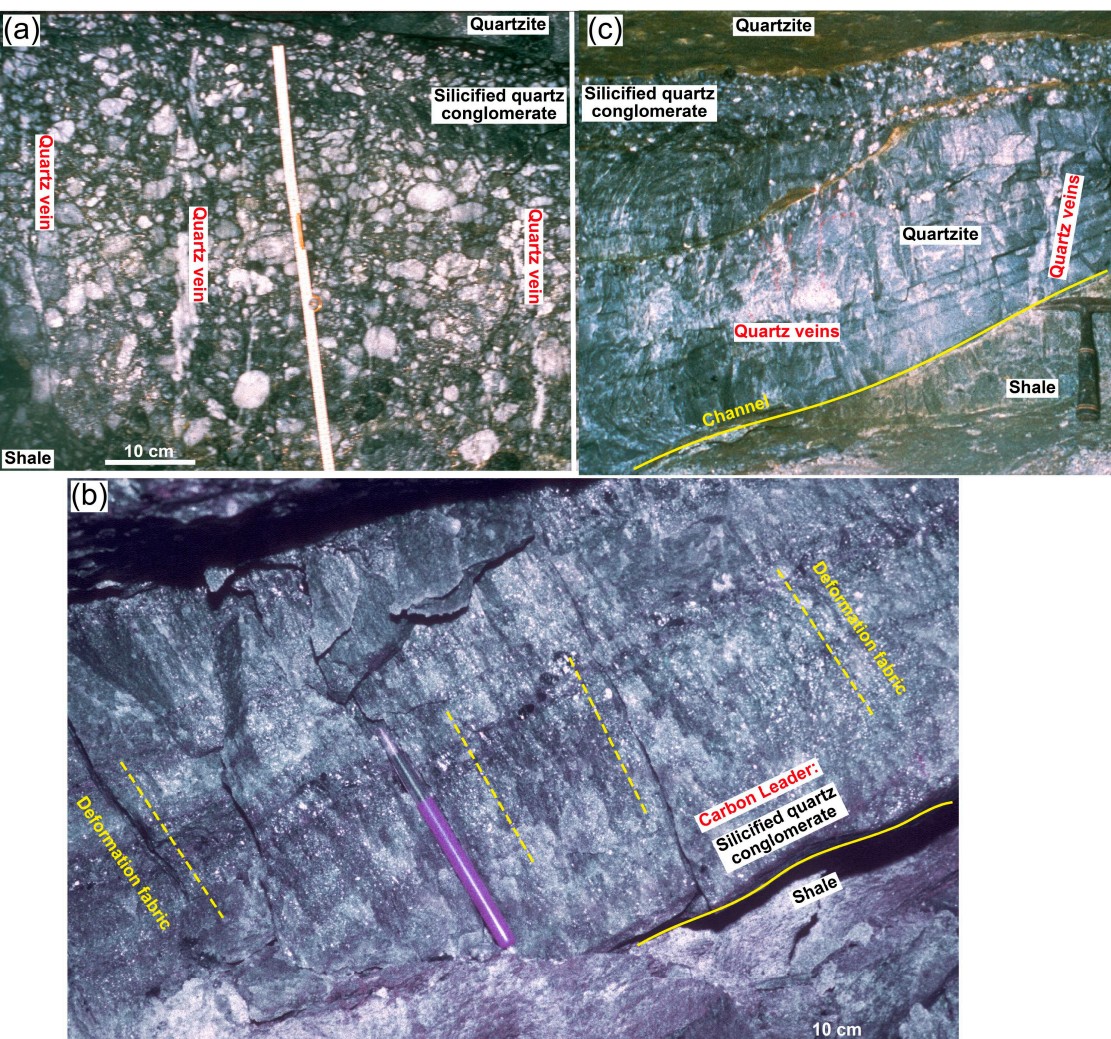

**Figure 5.** Photos of reef packages: (**a**) 4816 Ventersdorp Contact Reef, Kloof gold mine, Carletonville goldfield provided by Chamber of Mines Research Organisation. This is similar to the quartz pebble conglomerate, or banket, that was at the 1886 discovery site and typified the Central Rand goldfield and much early research [33]. The one metre metal tape measure is for approximate scale. Note the multiple subvertical quartz veins crossing most but not all of the conglomerate in the field of view. (**b**) 4620 Carbon Leader reef demarcated by the pen, with shale below the pen. Doornfontein gold mine, Carletonville goldfield. Carbon-bearing reefs such as this are very important in the Welkom to West Rand goldfields. In exploration globally, it is unlikely that any exploration approach developed to find conglomeratic banket ore (**a**) would recognise reefs such as this Carbon Leader. The photo has been interpreted as showing a fabric parallel to the pen defined by the long axis of quartz grains. (**c**) 4840 Kimberley Reef sand filled channel, Winkelhaak gold mine, Evander goldfield. The base of the channel is at the hammer pick, and quartz conglomerates are at the base of two higher erosion channels. Provided by Chamber of Mines Research Organisation.

### 4.2.2. Upper Witwatersrand (Central Rand Group)

The Central Rand Group (upper Witwatersrand) is almost 3 km thick and dominated by arenite, with thick conglomerate units more abundant towards the top (Figure 6). There is one 80 m thick shale unit (Booysens Shale) reflecting a significant marine interval. Referred to as 'quartzites' in the Witwatersrand literature, the arenites of the Central Rand Group are mostly sub-greywacke. The Group includes conglomerate, minor shallow shelf marine sandstone, and shale, but there is an absence of banded iron formations.

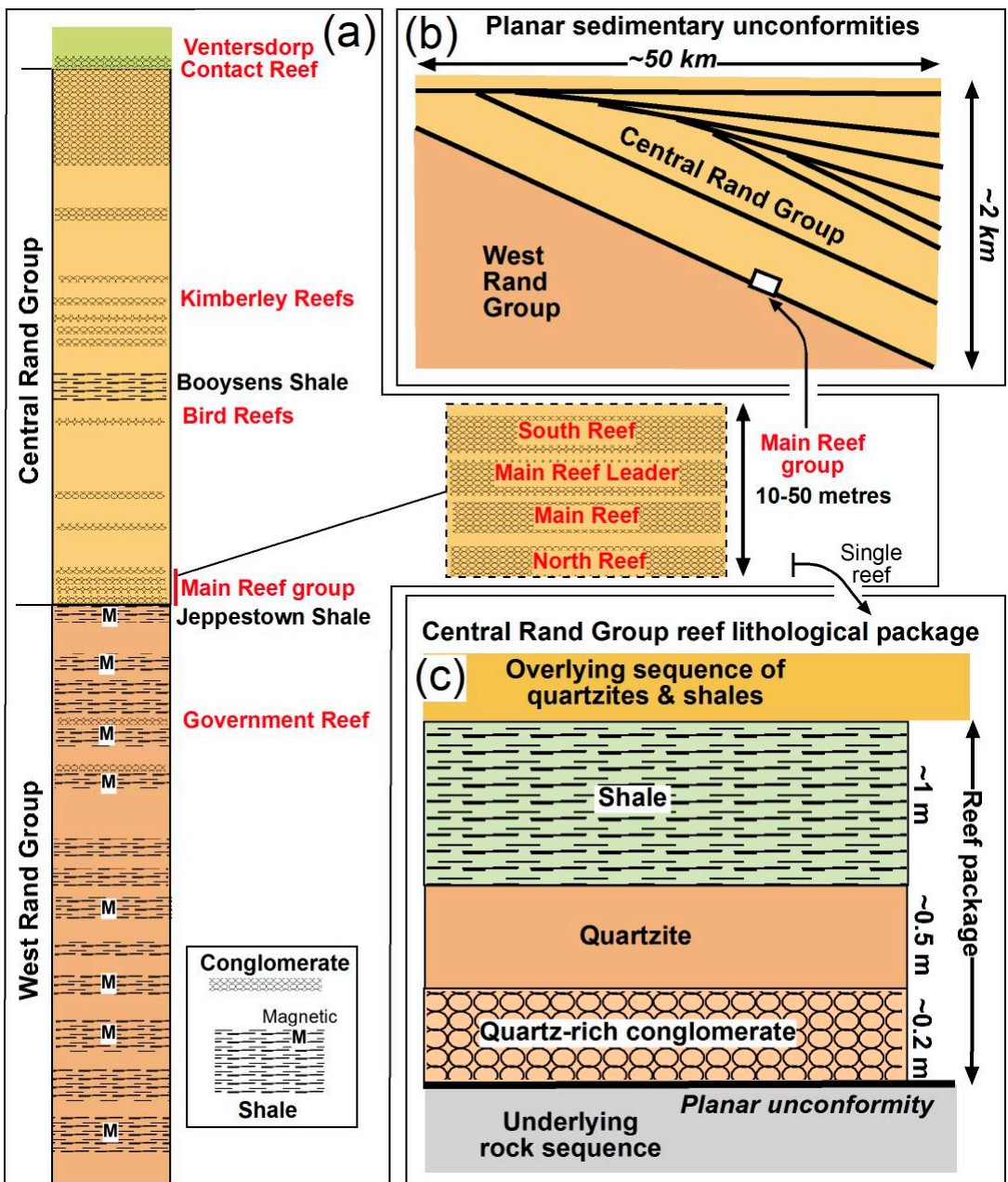

**Figure 6.** (**a**) Stratigraphic column of the Witwatersrand Supergroup, the position of some major reef groups, and the subdivision of those reef groups into reef packages. (**b**) Stacking of reef packages and unconformities, especially in the Central Rand Group. (**c**) A sketch of a reef package; these have contributed virtually all the gold production from the Witwatersrand.

In the middle part of the Central Rand Group (Booysens–Krugersdorp–Luipaardsvlei Formations), the age is best constrained as between 2815 ± 6 Ma (the youngest detrital xenotime grains calculated from U-Pb SHRIMP data [28]) and 2778 ± 3 Ma (the post-depositional xenotime data age [34]). This 2815 to 2778 Ma age bracket is for sediment deposition for this part of the Central Rand Group. This age window is further reduced to ~2790 to 2815 Ma if the recent baddeleyite ages for mafic sills intruding into the Witwatersrand Supergroup below the Central Rand Group are accepted to be feeders to the Ventersdorp Supergroup, immediately above the Central Rand Group [35].

The upper Witwatersrand crops out through Johannesburg and the West Rand, Klerksdorp, and Vredefort areas, but is also known from extensive underground mining operations. Because of the latter, it is easily the best-studied part of the supracrustal basins

being discussed here. It has an older Johannesburg Subgroup and a younger Turffontein Subgroup separated by the distinctive marker of the Booysens Shale, and each subgroup is ~1.5 km thick [9]. The dominant lithologies are sandstone and conglomerate with lesser shale. Some conglomerate units are up to a few metres thick and are reef packages like those of the lower Witwatersrand and Dominion Group. Other conglomerate units are tens to hundreds of metres thick, especially towards the top of the upper Witwatersrand. The banket type is famous for its gold production, such as that across the Central Rand goldfield [33]; the thick conglomerate units account for much less gold production.

*Reef packages* are the distinctive units from which most of the gold has been mined. They are composed of footwall rocks truncated against an unconformity overlain by carbon seam, thin conglomerate, clean sandstone, and shale, or some parts of this sequence ([24] figure 1). Reef packages have been described from the lower Witwatersrand (West Rand Group; [32]) and Black Reef but appear more abundant in the Central Rand Group [36]. The clean sandstone (quartzite) is very distinctive and given various names, including Marble Quartzite and Clean Bar. The reef packages are interpreted as marine transgression surfaces explaining their extent of hundreds of kilometres and planar nature [3].

*Reef group* is a term used to describe sets of closely spaced reef packages, with an example being the Main Reef group (lower case g is deliberate). It was within the Main Reef group that the first gold was discovered in 1886, and it comprises the North reef, Main reef, Main Reef Leader, and South reef, all within a thickness of 10–50 m in the Central Rand goldfield. The Main Reef group continues as a major producer to the East Rand, West Rand, and Carletonville goldfields. The Main Reef Leader, one of the reef packages in the Main Reef group, has produced 150 Moz (5000 t) Au on the Central Rand goldfield alone [37].

Four reef groups in the upper Witwatersrand stand out for their extraordinary gold production and each is mineralised on multiple goldfields (Figure 7). All have been correlated for at least 100 km by combining underground and drill core information, and they account for over 95% of all-time Witwatersrand production. From youngest to oldest, these are as follows:

- Ventersdorp Contact reef (VCR Elsburg)—4000 t Au (120 Moz);
- Kimberley—4000 t Au (120 Moz);
- Bird including Monarch, Vaal, Steyn, and Basal—15,000 t Au (500 Moz);
- Main Reef group, Carbon Leader, Nigel—28,000 t Au (900 Moz).

If these reef groups were thought of as four discrete gold deposits, they would be the first (Main Reef group), second (Bird Reef group), fourth, and fifth (Kimberley Reef group and Ventersdorp Contact Reef) largest gold deposits globally (Muruntau in Uzbekistan being the third). Next would follow Kalgoorlie and Grasberg.

Previous stratigraphic terminology mirrors these major reef groups, with the Main–Bird series and the Kimberley–Elsburg series, below and above the Booysens Shale, respectively. Several different rock types in these reef packages are the hosts for gold, especially carbon seams, oligomict conglomerate (i.e., banket), polymict conglomerate, and pyritic sandstone.

The sedimentology of the reef packages is as well-known as that of any Precambrian rock sequence, with thousands of measurements taken underground supplemented by core recording and field measurements. Thickness measurements define linear trends interpreted by some as channels in braided rivers and alluvial fans, and cross bedding and pebble measurements define water flow directions, depth, and sedimentary sorting regimes. An alternative interpretation is that the reef packages are heterogeneous rock sequences overlying marine unconformities [3,36], and this would account for their persistence over many 100 km$^2$, and the planar nature of gold mineralisation.

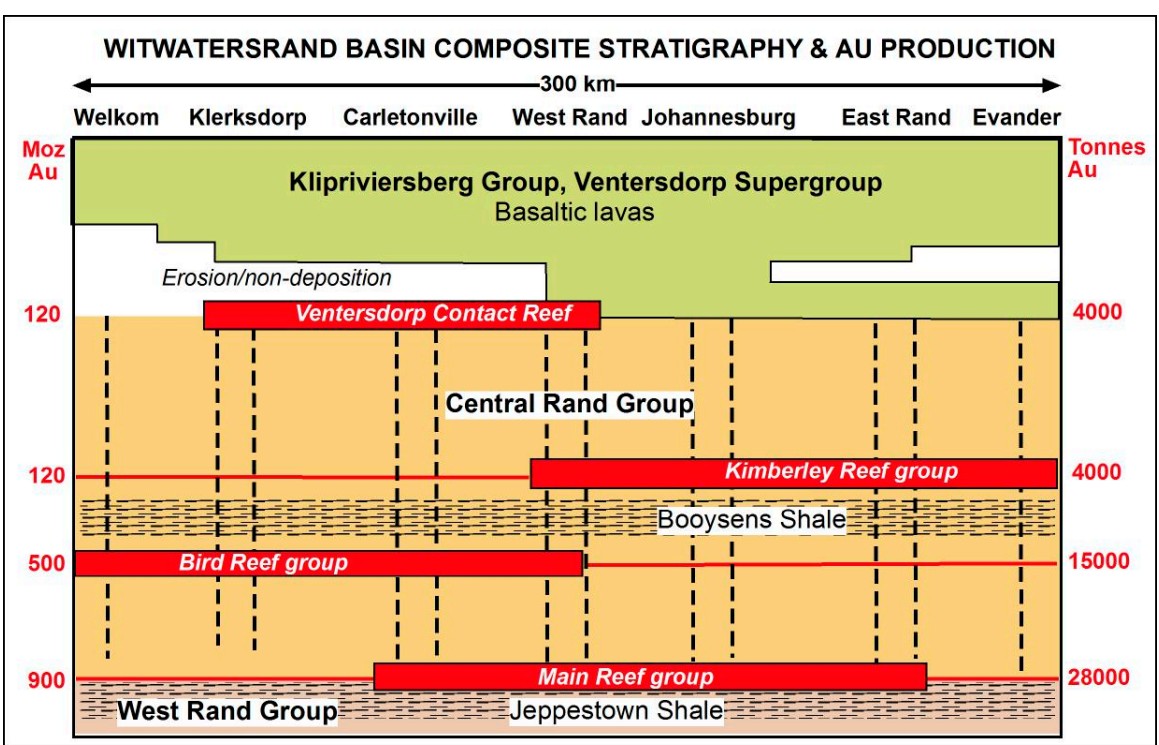

**Figure 7.** Unrolled long section through all the Witwatersrand goldfields from south (Welkom) to northwest to east (Evander as per Figure 1) illustrating the four major reef groups that have yielded over 1600 Moz Au (50,000 tonnes). Each reef group extends for 100 km or more and through multiple goldfields, and they can extend well beyond where they have been highly mineralised.

Separate from the sedimentological studies, detailed structural research within the Witwatersrand Supergroup has been focussed on reef packages from all of the goldfields and several different reef groups. These studies have revealed common bedding plane thrusting on the scale of centimetres to many metres, duplication of conglomerate by small ramps, and silty material that has lost silica through alteration and is commonly mapped as shale 'drapes' [38–41]. Thin quartz veining is abundant within metres of the reef packages in all goldfields, and minerals indicative of post-burial alteration (e.g., pyrophyllite, chloritoid) are widespread in all lithologies within and immediately footwall to the reef packages (except the clean quartzite). Although strain has been low and textures well-preserved in parts of the Witwatersrand, high-strain fabrics and thrusts are common [42,43] and occur in the reef packages with slate, phyllite, crenulated schist, and ultracataclastic fault rocks ([5], p. 309), [44,45].

Ventersdorp Contact Reef, Venterspost Conglomerate Formation

The Ventersdorp Contact Reef (VCR) has variously been considered the youngest major reef in the Witwatersrand Supergroup (the informal approach adopted here) or it is the base of the Ventersdorp Supergroup, and more formally given its own stratigraphic formation. The rationale for the formal classification is explained by [46] and based upon the supposed genetic differences between the VCR and typical Witwatersrand placers, reflected in differing mineralogical content and metamorphic grade: those supposed genetic differences are, in our opinion, rather problematic, but the assigned position of the VCR does not alter the direction and conclusions of this current paper.

Ventersdorp Supergroup

The Witwatersrand Supergroup is overlain by the Ventersdorp Supergroup conglomerate and tuff beds, some of which are dated. The basal Ventersdorp Conglomerate Formation is dated at 2729 ± 19 Ma [34]. And there is a slightly younger age on felsic rocks just above

the Ventersdorp Conglomerate Formation in the Klipriviersberg Group at 2714 ± 8 Ma (U-Pb SHRIMP [18]). The overlying Platberg Group of the Ventersdorp Supergroup is dated at 2709 ± 4 Ma [18], and less precisely 2693 ± 60 Ma [47] and 2643 ± 80 Ma [48].

The Ventersdorp Supergroup outcrops over 100 km, particularly west and south of the Witwatersrand Basin, and is known from drilling and geophysics over an even wider area [10]. It overlies the Witwatersrand Supergroup on an unconformity that is well-exposed in many gold mines and has a total thickness in excess of 4.5 km. The lower Klipriviersberg Subgroup comprises 2 km of lava, and the overlying Platberg Subgroup has a similar thickness of volcanic rocks and some clastic sedimentary rocks.

The Ventersdorp Supergroup is not noted for its auriferous conglomerates (excepting the importance of the underlying Ventersdorp Contact Reef), though there is anomalous gold near dykes and faults and within the lowermost ultramafic unit.

### 4.3. Black Reef Formation of the Transvaal Supergroup

Between the Ventersdorp Supergroup and the overlying Chuniespoort Group of the Transvaal Basin is the ~30 m thick Black Reef Formation of massive to cross-bedded quartz arenites, shales and siltstones, carbon rich shales, and minor conglomerates. The Black Reef Formation is not itself dated, although it is commonly described as 2590 Ma based on the relationship of rocks above and below [49] or 2642 ± 2 Ma [50,51], and we use 2640 Ma to 2590 Ma here. The Black Reef contains detrital zircons in age range from 3515 Ma mostly 3150 Ma to 3050 Ma, and to 2904 Ma as an overgrowth [51]. We are aware of no explanation for the >200 Ma gap between source and deposition, so it remains uncertain whether everything referred to as mineralised Black Reef is correctly the (basal) unit of the Transvaal Supergoup or not [49]. Importantly, the Black Reef is locally deformed, altered and gold mineralised (see below). Where the Black Reef has been mined, much of the Ventersdorp Supergroup is absent above the Witwatersrand Supergroup, allowing the Black Reef and Central Rand Group to be accessed in the same mines. Above the Black Reef lies the Oaktree Formation of the Chuniespoort Group (Transvaal Basin). Within this unit are tuff beds dated at 2583 ± 5 Ma and 2588 ± 7 Ma [52]. A supporting age on the Oaktree is 2550 ± 3 Ma [50].

Gold has been mined from the Transvaal Supergroup from the Pilgrim's Rest goldfields 300 km east of Johannesburg and this deposit is not discussed further, being so far away. Gold has also been mined from the Black Reef Formation from the locally auriferous and pyrite-rich Black Reef conglomerate, which is overlain by fluvial quartzite and a marine black shale [53,54]. The Black Reef Formation is reported as a ribbon around the 250,000 km$^2$ Transvaal Supergroup [54]. The Black Reef Formation has been thoroughly prospected along its hundreds of kilometres of outcrop, and the three major areas of gold mineralisation are in or adjacent to Witwatersrand goldfields ([55], p. 215), namely the Klerksdorp, West Rand, and East Rand goldfields. This relationship is confirmed by ([20], p. 186), who also noted that the Black Reef carries good gold grades only where in proximity to auriferous Witwatersrand reefs (noting that this means in proximity today rather than necessarily proximity during Archean deposition). Subsequently, ([14], p. 109) noted that "Black Reef conglomerates, always carry gold in close proximity to footwall Witwatersrand gold-bearing conglomerates".

Black Reef conglomerate has yielded gold from mines within the Klerksdorp goldfield (~0.3 Moz; [56]), from two about 30 km NE of Klerksdorp (~5 t Au), from Randfontein Estates on the West Rand (Lindrum Reefs, [57]), from four mines in the Central Rand goldfield less than 20 km SE of the centre of Johannesburg, and as subsidiary ores from major mines in the East Rand goldfields. On Consolidated Modderfontein mine in the East Rand, the Black Reef unconformably overlies the Witwatersrand Supergroup [58], and is mined on Modder Deep, Cons Modder, and Geduld in the East Rand [55]. Examples of the approximate size of these mines exploiting Black Reef ores include the Government Areas mine of the East Rand, which mined 30 Mt, the New Machavie mine 25 km north of Klerksdorp, which was stoped for a length of 1 km and to 150 m vertically to produce

5 t Au, and the Black Reef conglomerate near Klerksdorp, which was described as contributing handsomely to Klerksdorp production in 1895 [20]. Four mines SE of the centre of Johannesburg were small and produced only 0.2 Moz, and occurrences in the Carletonville goldfield appear to be small [51]. Carbon and uraninite are widespread in the Black Reef ores and commonly concentrated in upper parts of conglomerate rather than the base of channels ([55], p. 217).

Where an auriferous conglomerate mapped as Black Reef directly overlies the Witwatersrand Supergroup, as it does in some Witwatersrand mines, the correct stratigraphic position of that conglomerate reef can be problematic.

## 5. Synthesis of Gold Distribution

Important characteristics of the Dominion to Black Reef succession include the following:

- Auriferous conglomerates span a 400 my time span from the Dominion to Black Reef times (i.e., 3074 to 2640 Ma) and it is implausible that any special gold source remained accessible throughout this period but is seemingly absent today. Advocating erosion of older reefs to feed younger reefs does not explain gold in the pre-Central Rand Group rocks, nor does it make any difference to the ultimate placer requirement for 1500 Moz of detrital gold.
- The conglomerates formed in different tectonic settings, such as Dominion rifting, lower Witwatersrand thermal subsidence, upper Witwatersrand foreland basin, Ventersdorp flood basalt province and later Ventersdorp graben, and early Transvaal thermal subsidence ([13,15] and references therein). It is implausible that the enormous source of detrital gold continued to be available for erosion in all these settings. There is also no support for special sedimentary sorting processes in a wide heterogenous and diverse range of sequence settings.
- Despite the relevant basins having dimensions of hundreds of kilometres, all the sustainable mining operations mentioned are in or adjacent to the main upper Witwatersrand goldfields.
- The host rocks for most of the economic gold are quite varied, noting that the focus here has been on conglomerate. The main host rock association is with carbon (i.e., carbon seam), e.g., Basal Reef, Vaal Reef, Carbon Leader Reef, and not the conglomerate per se. The important host rocks are carbon seams, oligomict conglomerate locally called banket in the past, polymict conglomerate, and pyritic sandstone [13]. Differing sedimentary rocks such as these reflect differing sedimentary depositional environments and processes, and most likely different source regions, and do not indicate the special source and sorting needed for a major placer deposit.
- All the rocks have been deformed and metamorphosed, including a widespread overprint of pyrophyllite alteration (Figures 8 and 9).

In Phanerozoic terms, this is equivalent to a static source region for gold and consistent special sorting processes in a limited location from the Devonian to the recent. This is a period sufficient for a reconfiguration of the planet's continents in two major orogenic cycles. The concept of preserving a local, yet extremely effective, source and sorting process for this length of time is contrary to what we know from the Phanerozoic and has no support in the Precambrian record of South Africa.

The Klerksdorp district is especially enlightening, with outcrops of auriferous conglomerates in the Dominion, lower Witwatersrand, upper Witwatersrand, Ventersdorp Contact Reef, and Black Reef. Given the lack of outcrops at most of the other goldfields, it may be that the Klerksdorp pattern of gold distribution is the unrecognised norm that has been undetected elsewhere due to lack of access to older parts of the sequence.

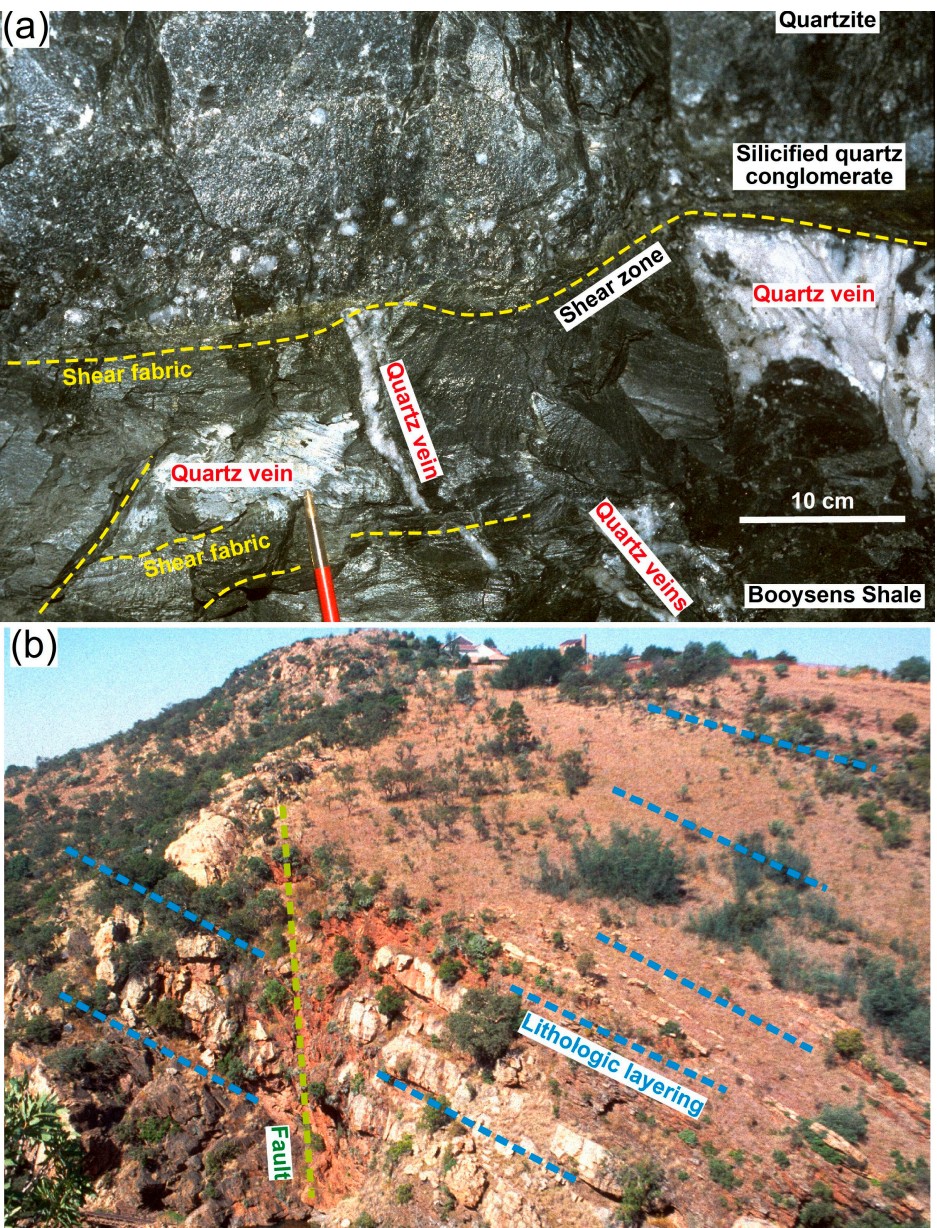

**Figure 8.** Deformation features in underground and on surface. (**a**) 4625 Kimberley Reef (horizontal, central) with small quartz pebbles, overlain by clean quartzite, and underlain unconformably by Booysens Shale with some quartz veins that appear to terminate at the change of rheology. The base (mid-left) of the conglomerate shows a distinct deformation cleavage. Grootvlei gold mine, East Rand goldfield, #6 shaft, 6 level. (**b**) 4280 View of the basal parts of the Witwatersrand Supergroup near the suburb of Roodepoort, Johannesburg. This is looking east with outcropping white sandstone units at a moderate dip to the south (right). The quartzites form several such east–west ridges through Johannesburg and give the *Witwatersrand* name (white water ridges). The red rubbly area is a steep fault that has disaggregated the quartzite units, allowing access to groundwaters and giving red Fe-oxide and clay soils. This is approximately 20 km NW of the 1886 discovery site and central Johannesburg.

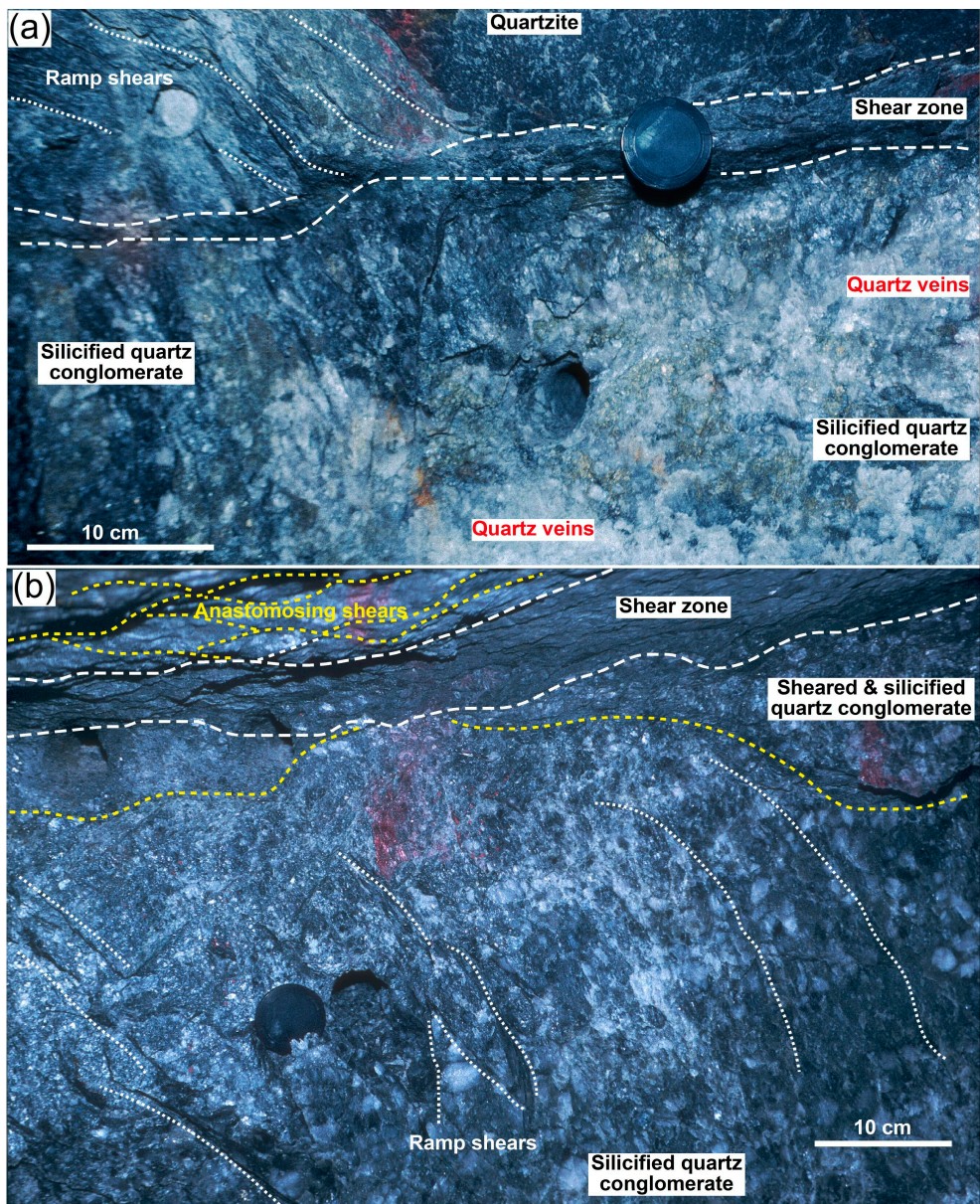

**Figure 9.** Kimberley reef, Marievale gold mine highlighting variability, quartz veins, and high strain zones. (**a**) 4753 High grade Kimberley reef (3 cm thick, approximately horizontal through lens cap), shear zone in clean quartzite hanging wall, quartz vein in footwall to the reef, and carbon near the top of the conglomerate. Eight level, Marievale mine, East Rand goldfield. (**b**) 4756 High-grade Kimberley reef of quartz pebble conglomerate below strongly foliated pyrophyllite-bearing schist (upper quarter of photo), 8 level, Marievale mine, East Rand goldfield.

## 6. Implications for Exploration Models

The placer model for Witwatersrand gold has been a foundation for most exploration for more than half a century. Implicit in the model is the hypothesis that there has existed an enormous source for detrital gold for which there is no realistic candidate, and that there has been very special sedimentary sorting to concentrate grains of gold, which is not supported by the diversity of host rocks for the gold. The broad scale distribution of economic gold, as documented here, is contrary to placer processes and makes a special source or a special sorting process unlikely. A geological model assuming placer processes that would require special source and sorting parameters that persisted through such a long time and stratigraphic interval, and through variable tectonic environments, to produce the

world's largest gold deposit, is implausible. These shortcomings create opportunities for new exploration models that might be more successful.

Future Witwatersrand gold exploration could consider these findings, and the large expenditure already absorbed by placer-based exploration; the lack of success finding a new goldfield since 1951 should be a scientific and commercial "alarm bell" or "wake up call" to indicate that funds might be better invested using other genetic models [2]. We suggest that instead of continual special pleading for the placer model with further and further modifications as new conflicting information emerges, a more scientific approach might be to consider alternative models based upon a broader understanding of the formation of many gold deposits globally, and not to base so much exploration and research around the placer model.

The present synthesis indicates the value of adding a vertical dimension into an exploration model and paying more attention to gold distribution patterns orthogonal to layering (beyond simply analysing every conglomerate). The lateral focus along conglomerate horizons and unconformities, which is part of the placer and most other exploration models, is useful but incomplete.

Structural geology might be incorporated more thoroughly into Witwatersrand exploration beyond the very important seismic interpretations of stratigraphic geometry. Strain in the reef packages is greater than in many units of the Witwatersrand Supergroup; it is heterogeneous and associated with concentrations of quartz veins and hydrothermal alteration with pyrophyllite [40,59]. All of this indicates that the reef packages were likely fluid channel ways. Rheology and deformation styles in these sedimentary sequences may be very informative and strongly influenced by bedding-parallel features and heterogeneous rock packages in otherwise thick uniform sequences, i.e., some reef packages might be considered as potential high strain zones of fluid flow.

A hypothetical example demonstrates the futility of a universal placer model for Witwatersrand gold. If, after discovering the Main Reef in the Central Rand, one was to take a descriptive approach and follow similar quartz pebble conglomerates, new mineralisation would be found. The addition of a genetic approach using sophisticated sedimentology would (and has) not added anything of *predictive* value, such as why to look in a polymict conglomerate or pyritic sandstone. Taking this example further, there would be no logical reason to look kilometres below or above the Main Reef for more auriferous conglomerates in other supergroups. And, a lateral search beyond the Main Reef in the Central Rand goldfield based upon descriptive similarities and sedimentology would not find the Carbon Leader, Basal Reef, or Vaal Reef. These reefs were discovered using geophysics, deep drilling, and stratigraphy, including the geology of critical unconformities.

An exploration model that incorporates geochemistry and the gold precipitating capacity of Fe (now pyrite, pyrrhotite, and arsenopyrite) and carbon would draw exploration attention to the potential for gold in these carbon-bearing horizons within this hypothetical example just mentioned. This exploration approach could be enhanced by incorporating the gold-related alteration, including pyrite as an envelope around mineralisation and as a guide to fluid flow channelways. All of this might require training teams about hydrothermal processes, complexation and the role of ligands, and the chemistry of carbon as a metal precipitant and its mode of formation [60].

A geological model using hydrothermal principles, in different forms, has been suggested for Witwatersrand gold for a century, but this does not mean that it has been seriously applied in large-scale exploration. References to the hydrothermal model by most authors are typically to demonstrate that the model is not correct, and this is hardly a platform from which to launch a successful exploration program. A different approach might be to recognise that there is an opportunity to use ideas apart from the placer model. These would be supported by learning all aspects of a hydrothermal model, then training the exploration team, and including management so that they fully understand what they are being asked to fund.

### 7. Summary and Conclusions

Very few ancient conglomerates have provided ore for large-scale and long-term (i.e., sustainable) gold mining operations, and most of these are in northern South Africa and within the Dominion Group, lower and upper Witwatersrand groups, and Black Reef Formation [16]. Considering these occurrences in combination, it is difficult to explain this broad gold distribution in terms of a geological model that assumes placer processes for the gold, or any other syngenetic model. There is no trace of the necessary enormous source of detrital gold spanning a period of 400 my nor of the necessary and very effective sedimentary sorting process. The model does not explain the economic gold encountered in a variety of hosts, including carbon-rich units, pyritic quartzites, and reef package shales, nor anomalous gold in rocks adjacent to cross-cutting channelways, such as dykes and faults. Without a viable gold source and special sedimentary sorting process, the placer model is a poor basis for any Witwatersrand gold exploration.

This re-examination of mines around the Witwatersrand Basin provides a new perspective on gold distribution and, in turn, provides new opportunities for exploration. The gold distribution might be better modelled by considering hydrothermal processes. The benefit of using a different (hydrothermal) model is that it has hardly been applied in the Witwatersrand with serious intent by teams familiar with discovery, *despite the same hydrothermal model having been applied with great success globally* in a variety of Archean and younger terrains. Just like the discovery of Carletonville, a goldfield discovery using this new approach may generate successes in unusual settings and well beyond initial expectations.

**Author Contributions:** Underground research on each gold mine, N.P.; South African geological components, N.P. and J.V.; figures, D.C.; manuscript development, D.C., N.P., J.V. and A.D. All authors have read and agreed to the published version of the manuscript.

**Funding:** This research received no external funding.

**Data Availability Statement:** Dataset available on request from the authors.

**Acknowledgments:** Two underground photos (4820, 4840) were provided by the Chamber of Mines Research Organisation, Matt Mullins and other COMRO collaborators. Other underground photos are from the Phillipsgold collection. Comments from Jun Cowan on the manuscript and discussions with Andy Barnicoat are greatly appreciated.

**Conflicts of Interest:** Julian Vearncombe is an employee of Effective Investments Pty Ltd. The paper reflects the views of the scientists and not the company. The authors declare no conflict of interests.

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
