# Peer review of "The Temporal Distribution of the Host Rocks to Gold, the Archean Witwatersrand Basin, South Africa"

_minerals, doi:10.3390/min14020199_

Round 1

Reviewer 1 Report

Comments and Suggestions for Authors

The manuscript by Phillips et al. entitled "The temporal distribution of the host rocks to gold, the Archean Witwatersrand Basin, South Africa" presents a new look at the famous "Wits" through the lens of the host rocks to the gold.

It was a pleasure to read this manuscript - novel, interesting and overall well structured and well written.

I recommend that this manuscript be accepted for publication in Minerals after minor revisions taking into account my comments in the attached PDF.

Best regards,

A reviewer

Reviewer 2 Report

Comments and Suggestions for Authors

The long-lasting controversy as regards the singenetic vs. epigenetic (or placer vs. hydrothermal) character of the Witwatersrand Basin gold mineralization appears to have settled in recent years towards a “modified placer model”. This convergence resulted from large volume of sedimentological, geochemical, isotopic and textural evidence accumulated in decades of multilateral studies, not without a tight connection with the mentioned long-lived controversy, besides knowledge of the geometry of the mineralization resulting from the mining activity. Illustrative in this respect are more or less extensive syntheses (Frimmel et al., 2005 – a paper totally ignored throughout the manuscript; Tucker et al., 2016, etc.) reviewing and weighing the data available.

In this context, the manuscript submitted places itself at the far end of the spectrum, claiming a purely hydrothermal origin of the deposits and consequently arguing for a change in the exploration strategies, which should, in the opinion of the authors, be based on epigenetic models. In doing so, the general perspective appears biased, and the exposition is excessively assertive. The title of the manuscript makes explicit reference to the age of the host rocks of the mineralization, topic which is briefly treated in ch. 4,  (Geological setting of reef packages and reef groups) by a reference to Dankert and Hein (2010) “used in simplified form”. The declared aim of the manuscript is to synthesize (once again) data regarding the age, host package, stratigraphic position, tectonic setting at sedimentation, geographic setting today, deformation and relationship to large-scale alteration of economic gold in Archean conglomerates, in order to answer the question whether a geological model assuming a placer origin remains fit for purpose today. As a starting point, the authors chose the contribution of Pretorius (1981) somewhat surprisingly, given the more recent syntheses mentioned, and further on focus on two aspects: assessing “the value of Pretorius’s syntheses and judgements” and challenging the detrital deposition of gold in favour of a hydrothermal origin of the mineralization, without entering in detail or supporting it with pertinent arguments. It is rather an attempt to discredit the detrital source model based on doubting “plausibility” because of the long depositional span allegedly in different tectonic settings (counting also the Dominion Group and the Ventersdorp Supergroup, or even the Black Reef belonging to the Transvaal Supergroup, for convenience). A distorted image of gold distribution is conveyed by ignoring the concentration of the most productive reefs almost exclusively in the Central Rand Group. Other assertions prone to induce a distorted view consist in mixing anomalous contents in certain rock types with economic-grade mineralization, claiming a diversity of host rocks without providing at least approximate quantitative data, discussing low-strain domains and sedimentary structures preserved “in parts” versus high-strain fabrics and intraformational thrusts which “are common”, without giving a quantitative estimate of their size and ratio.

There is no attempt whatsoever to discuss concrete evidence published regarding sedimentary structures and ore distribution (eg. Tucker & Viljoen, 1986), detrital origin of part of the ore grains based on morphology and granulometry (eg. Vollbrecht et al., 2002), pre-depositional ages (eg. Kirk et al., 2002), inner structure (Fleet, 1998), geochemical features and many other supportive details.

The hydrothermal model invoked is not exposed in detail as regards its source and timing; apparently it adds to the problem of a source for so much gold, that of an effective pathway in compacted rocks. Its application and the benefits for exploration remain unclear from the text of the manuscript.

I do not recommend the publication of the manuscript in the present form, a major revision should be carried out in order to add reasonable and detailed arguments for contesting the invoked evidence accumulated in support of a detrital origin of the mineralization.

Notable is the dissatisfaction expressed by the authors: “References to the hydrothermal model by most authors is typically to demonstrate that the model is not correct, and this is hardly a platform from which to launch a successful exploration program.”, while doing exactly the same thing.
